# Effect of Deadwood Decomposition on the Restoration of Soil Cover in Landslide Areas of the Karpaty Mountains, Poland

**Wojciech Piaszczyk** [1,*] **, Jarosław Lasota** [1]**, Grzegorz Gaura** [1,2] **and Ewa Błońska** [1]

1   Department of Ecology and Silviculture, Faculty of Forestry, University of Agriculture in Krakow,
    29 Listopada 46 Str., 31-425 Kraków, Poland; jaroslaw.lasota@urk.edu.pl (J.L.);
    grzegorz.gaura@gorcepn.pl (G.G.); ewa.blonska@urk.edu.pl (E.B.)
2   Gorce National Park, Poręba Wielka 590, 34-735 Niedźwiedź, Poland
*   Correspondence: wojciech.piaszczyk@urk.edu.pl

**Abstract:** Disturbances play an essential role in the shaping of temporal and spatial heterogeneity in natural community structures. The aim of this study was to provide an assessment of the deadwood influence on the chemical and biochemical properties of soils in a landslide area. The samples used to determine soil properties were collected from the entire landslide area, with locations distributed on a regular grid (50 × 50 m). The soil samples were collected from directly under the logs, and background soil samples were taken 1 m from the deadwood logs. The effect of the deadwood decomposition process was visible in the total organic carbon (C) and nitrogen (N) content and microbial activity of the soil. An increase in the enzyme activity and microbial biomass of the soil from directly beneath the deadwood was noted. In this study, it was found that a greater stock of deadwood was present in the accumulation zone, which resulted in a stronger effect of the released components on the soil cover. In order to restore landslide soils, microbial activity can be effectively stimulated by leaving deadwood on the landslide surface.

**Keywords:** enzyme activity; humus mineral horizon; microbial biomass



## 1. Introduction

"Landslide" is the term used to describe rocky masses dislocated from higher ground and displaced downslope [1,2]. According to Haque et al. [3], $3.7 \times 106$ km$^2$ of land surface is prone to landslides worldwide. Landslides are a common phenomenon in mountainous areas. One region in the Carpathians is the most threatened in Poland. The Carpathians cover 190 thousand km$^2$, and in Poland 19.6 thousand km$^2$. This area constitutes only a few percent of the country's area, but over 90% of all landslides in Poland are located in this region. Landslides are mainly favored by the terrain and the appropriate type, arrangement and depth of rock formations [4]. Such geological phenomena often occur in areas dominated by flysch rocks. Most often, landslides take place on slopes with a slope of 10–20°. Mechanically, they occur when alternating beds of sandstone and claystone, separated by shale or clay/slate, are inclined obliquely or parallel to the slope, facilitating movement [5]. The factors contributing to the formation of landslides are intense and long-lasting rainfall and human activity. The recovery of soil and vegetation after a landslide requires time, during which the landslide area is exposed to erosion processes and the movement of rock and soil material [6]. As a result of landslide movements and the mixing of soil materials, soil properties change dramatically [7,8]. The greatest differences in properties have been recorded in landslide niche soils when compared to the area of material accumulation (landslide foot) and the edges of the landslide [9].

As a result of landslides, soil cover and vegetation are significantly or completely destroyed, while the biomass of damaged, dead trees remains on the surface of the landslide. Deadwood is a source of organic matter and nutrients that are transferred into the soil [10,11]. Deadwood thus modifies the physical, chemical and biochemical properties of

soils under its direct influence [12,13]. Several different forms of nutrients are released from deadwood. Ammonium and nitrate ions, and phosphorus in the form of phosphate ions, are released into the soil [12], and these influence the enzyme activity in the soils beneath the logs [11]. Changes in the quantitative and qualitative structures of soil microorganisms and their biological activity have been observed in such soils [14,15]. The microbiological and biochemical properties of soils have been used to assess soil condition to great effect [16], with the biochemical characteristics being related to the amount and activity of soil microorganisms [17]. Soil enzymatic activities are sensitive bioindicators of all natural or anthropogenic disturbances [18]. Soil enzymes play an important role in soil organic carbon (C) mineralization (β-D-cellobiosidase, β-glucosidase, β-xylosidase), nitrogen (N) cycle (N-acetyl-β-glucominidase), phosphorus (P) cycle (phosphatase) and sulphur (S) cycle (arylsulphatase) [19].

To date, studies have focused on determining areas predisposed to landslides, with the majority of these focusing on the tropical climate zone [6,20,21]. The ecological role and importance of this kind of disturbance are, however, seldom taken into account. It is well known that disturbances play an important role in nature, providing a major source of temporal and spatial heterogeneity in natural community structures and dynamics [22]. Earlier studies focused on landslide soils, but did not include the effect of deadwood on the properties of soil cover [9,23]. This study addresses this knowledge gap by assessing the recovery of landslide-related soils and the role of deadwood in the soil recovery process. It was expected that the deadwood would improve the properties of the soil cover on the landslide, regardless of the zone, especially C and N content. It was also assumed that the nutrients released from the deadwood, especially C substrates, would stimulate the enzyme activity and microbial biomass of soils on the landslide.

## 2. Materials and Methods

### 2.1. Sampling Sites and Experimental Design

The research was carried out in the area of the Sucha Forest District (southern Poland). The study covered a landslide that took place in July 2001. The landslide covered an area of about 5 km$^2$ (500 m long and 100 m wide). Information about the landslide was obtained from the Polish Institute of Geology and was confirmed by the State Forest staff. The landslide was located at an altitude of 550–660 m above sea level (a.s.l.), and the sampled plots occupied an area underlain by the Cenozoic Magura Sandstone Formation and a schist facies. Dystric Cambisols prevail in the test area [24] and silver fir (*Abies alba*) was the dominant species in the area before the landslide. The landslide area was without trees, the new generation of fir appeared sporadic, and *Rubus hirtus* was dominant in undergrowth vegetation. There was no new generation of trees on deadwood and in the immediate vicinity. The study area is characterized by the following climate conditions: the average annual rainfall is 940 mm, the average annual temperature is 7.2 °C, and the length of the growing season is 213 days [17]. Soils of the landslide were characterized by a similar texture; the average sand content was 45%, silt content was 49% and clay content was 6%.

Based on the morphology of the area, zones of depletion and accumulation were determined on the landslide (Figure 1). Both zones were characterized by a similar slope inclination (30–35 °C). The soil samples to determine basic properties were collected from the entire landslide area, using points distributed on a regular grid of 50 × 50 m, yielding 20 samples (Figure 1). Soil samples were collected from directly beneath logs of wood using a small spade. Samples of background soils were taken 1 m from the logs. In previous studies, we proved that there was no influence of decaying wood on the soil at a distance of 1 m from the log [25]. All of the soil samples were collected from the humus mineral horizon A (0–10 cm). Soil samples for laboratory analysis were taken in September 2018.

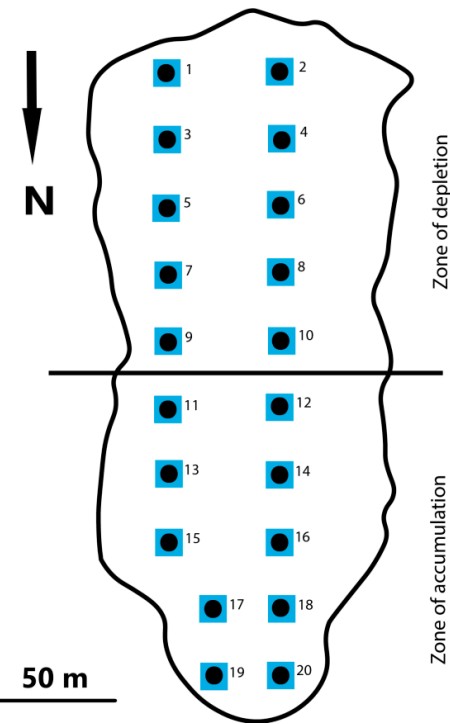

**Figure 1.** Soil sampling scheme on landslide area.

The stock and decay class of the deadwood was inventoried on the landslide. The Maser et al. [26] classification was used to estimate the degree of decay. The volume of the deadwood was calculated according to the formula below [27]:

$$V = (\pi\,d^2)/40{,}000\ L$$

where V is volume in $m^3$, L is length (of lying dead trees) and d is diameter at half the length for logs and diameter at the base for stumps.

The most decomposed (fifth degree) fir logs were selected for a detailed examination of the impact of deadwood on soil properties, after an analysis of the amount of deadwood and its degree of decomposition.

### 2.2. Laboratory Analysis

The pH of the samples was measured potentiometrically in $H_2O$ and KCl. The base cations ($Ca^{2+}$, $Mg^{2+}$, $K^+$, $Na^+$) were determined after extraction in 1M ammonium acetate by inductively coupled plasma−optical emission spectrometry (iCAP6500 DUO, Thermo Fisher Scientific, Cambridge, UK). The N and C contents were also determined (Leco CNS 2000 True Mac Analyzer, St. Joseph, MI, USA), and the C/N ratio was calculated. The particle-size distribution was determined using laser diffraction (Analysette 22, Fritsch, Idar-Oberstein, Germany). The microbial biomass C, N and P were determined, using the fumigation and extraction method [28–30]. The enzyme activity (β-D-cellobiosidase (CB), β-xylosidase (XYL), N-acetyl-β-D-glucosaminidase (NAG), β-glucosidase (BG), phosphatase (PH) and arylsulphatase (SP)) was determined using fluorogenically labeled substrates [31–33]. The fluorescence was measured on incubated soil suspensions (1.5 h at 35 °C) in 96-well microplates (Puregrade, Germany), with the fluorescence determined immediately on a multi-detection plate reader (SpectroMax, Molecular Devices), with excitations set at 355 nm and emission at 460 nm. All analyses were performed in three repetitions. All analyses were performed in the certified Geochemistry Laboratory of Environment and Forest Areas Designed for Reclamation (certificate of the Polish Center of Accreditation AB1656). Certified Reference Material (CRM) was used during the analyses.

### 2.3. Statistical Analysis

The Manna–Whitney U test was used to determine the differences between the mean values of the soil properties. Pearson's correlation coefficient was calculated for the biochemical soil properties and the basic chemical characteristics. Principal component analysis (PCA) was employed to identify interrelationships among the variables and to identify how suites of variables were related. The relationships between the soil and deadwood properties in the two zones of the landslide were evaluated using PCA. We adopted a general linear model (GLM) to investigate the effect of the deadwood occurrence and localization on landslide area (zone of depletion or zone of accumulation) on the carbon content and enzyme activity of soil. Differences with $p < 0.05$ were considered to be statistically significant. All analyses were performed using Statistica 12 software (StatSoft 2012).

## 3. Results

The results of the deadwood inventory on the landslide indicated that a greater amount was in the accumulation zone (370.76 m$^3$ ha$^{-1}$). In both zones, most of the deadwood was in decay classes IV and III, with less in class V. There was no wood in decay class I or II on the landslide surface (Table 1).

**Table 1.** Stock of deadwood in different decay classes (m$^3$ ha$^{-1}$) on landslide area taking into account the division into zones.

| DC | Zone of Depletion | Zone of Accumulation |
|---|---|---|
| I | 0 | 0 |
| II | 0 | 0 |
| III | 92.12 | 131.51 |
| IV | 100.00 | 143.81 |
| V | 64.04 | 95.44 |
| Sum | 256.16 | 370.76 |

DC—decay classes.

The influence of decaying wood on the soil properties was recorded in both landslide zones (Tables 2 and 3). Statistically significant differences in C and N content were found between samples under the influence of deadwood and samples from the background, regardless of landslide zone (Table 2). High C and N contents were determined in the soil affected by deadwood in both landslide zones. The mean C content in samples influenced by deadwood was 4.77% in the depletion zone and 6.61% in the soil accumulation zone, while in the background samples it was 3.07% in the depletion zone and 4.08% in the accumulation zone. The mean N content in the samples under deadwood in the depletion zone was 0.37%, while in the accumulation zone it was 0.29%. In the background samples, it was 0.25% in the depletion zone and 0.19% in the accumulation zone. The pH was also high in the soil samples influenced by deadwood, regardless of the landslide zone (Table 2). There were no statistically significant differences in the Ca, Na, K and Mg contents between the samples, regardless of landslide zone. There were also no statistically significant differences in the microbial biomass C and N, although statistically significant differences in the microbial biomass P were observed. The highest value of microbial biomass C, N and P was found in those samples affected by deadwood, regardless of landslide zone (Table 3). The average values of enzymatic activity of all tested enzymes were higher in the soil samples affected by deadwood than in those from the background, regardless of landslide zone (Table 3). Statistically significant differences in all enzyme activity were found between samples under deadwood influence and samples from the background (Table 3).

**Table 2.** Chemical properties of soil under deadwood influence and background samples on landslide area taking into account the division into zones.

| Type of Soil Samples | | N | C | C/N | pH H$_2$O | pH KCl | Ca | Na | K | Mg |
|---|---|---|---|---|---|---|---|---|---|---|
| Zone of depletion | Soil under deadwood | 0.37 ± 0.22 [a] | 4.77 ± 1.96 [a] | 17.64 ± 3.67 [a] | 4.98 ± 0.65 [a] | 4.31 ± 0.76 [a] | 3.72 ± 0.88 [a] | 0.04 ± 0.01 [a] | 0.18 ± 0.05 [a] | 0.58 ± 0.31 [a] |
| | Background | 0.25 ± 0.14 [a] | 3.07 ± 1.09 [b] | 16.01 ± 1.22 [a] | 4.62 ± 0.55 [a] | 3.69 ± 0.32 [a] | 3.48 ± 1.01 [a] | 0.03 ± 0.01 [a] | 0.19 ± 0.06 [a] | 0.51 ± 0.26 [a] |
| Zone of accumualtion | Soil under deadwood | 0.29 ± 0.10 [a] | 6.61 ± 4.57 [a] | 16.17 ± 3.57 [a] | 5.21 ± 0.55 [a] | 4.59 ± 0.57 [a] | 4.24 ± 1.14 [a] | 0.04 ± 0.01 [a] | 0.20 ± 0.07 [a] | 0.72 ± 0.24 [a] |
| | Background | 0.19 ± 0.06 [b] | 4.08 ± 2.37 [a] | 15.61 ± 1.57 [a] | 4.86 ± 0.40 [b] | 3.88 ± 0.30 [a] | 3.44 ± 0.96 [a] | 0.04 ± 0.01 [a] | 0.22 ± 0.14 [a] | 0.65 ± 0.31 [a] |

C—organic carbon content (%), N—nitrogen content (%); Ca, K, Mg and Na (cmol(+)·kg$^{-1}$); different lowercase letters in the upper index ([a,b]) mean significant differences of parameters between soil under influence of deadwood and that without influence of deadwood.

**Table 3.** Microbial biomass and enzyme activity of soil under deadwood influence and background samples on landslide area taking into account the division into zones.

| Type of Soil Samples | | MBC | MBN | MBP | MBC/MBN | MBC/MBP | MBN/MBP | CB | BG | NAG | XYL | SP | PH |
|---|---|---|---|---|---|---|---|---|---|---|---|---|---|
| Zone of depletion | Soil under deadwood | 937.36 ± 422.59 [a] | 211.13 ± 179.15 [a] | 4.64 ± 6.13 [a] | 5.96 ± 4.20 [a] | 330.39 ± 139.29 [a] | 68.75 ± 42.03 [a] | 40.76 ± 37.79 [a] | 163.17 ± 103.85 [a] | 235.23 ± 116.48 [a] | 89.92 ± 59.32 [a] | 40.93 ± 37.88 [a] | 1368.29 ± 917.15 [a] |
| | Background | 572.23 ± 102.87 [a] | 110.42 ± 81.01 [a] | 1.58 ± 1.12 [a] | 9.40 ± 8.91 [a] | 442.23 ± 178.97 [a] | 88.59 ± 36.76 [a] | 17.43 ± 17.47 [a] | 108.66 ± 76.01 [a] | 119.51 ± 86.66 [b] | 20.21 ± 11.75 [b] | 13.38 ± 7.62 [b] | 776.16 ± 510.73 [a] |
| Zone of accumualtion | Soil under deadwood | 897.32 ± 335.51 [a] | 118.92 ± 102.08 [a] | 2.31 ± 0.94 [a] | 5.77 ± 2.02 [a] | 418.42 ± 169.18 [a] | 80.26 ± 36.48 [a] | 69.22 ± 32.01 [a] | 226.39 ± 62.07 [a] | 228.43 ± 149.31 [a] | 65.96 ± 36.63 [a] | 51.07 ± 51.05 [a] | 1496.65 ± 400.56 [a] |
| | Background | 612.13 ± 273.61 [a] | 113.54 ± 82.98 [a] | 1.54 ± 0.87 [b] | 8.83 ± 7.00 [a] | 455.63 ± 289.10 [a] | 81.11 ± 63.75 [a] | 33.06 ± 27.46 [b] | 147.21 ± 61.25 [b] | 112.61 ± 22.35 [b] | 23.55 ± 17.35 [b] | 19.05 ± 12.59 [a] | 1073.85 ± 224.11 [b] |

MBC—microbial biomass carbon (µg·g$^{-1}$), MBN—microbial biomass nitrogen (µg·g$^{-1}$), MBP—microbial biomass phosphorus (µg·g$^{-1}$); CB—β-D-cellobiosidase, XYL—β-xylosidase, NAG—N-acetyl-β-D-glucosaminidase, BG—β-glucosidase, PH—phosphatase and SP—arylsulphatase (nmolMUB·g$^{-1}$ d.s.·h$^{-1}$); different lowercase letters in the upper index ([a,b]) mean significant differences of parameters between soil under influence of deadwood and that without influence of deadwood.

A positive correlation was found between the C and N content, and between the enzymatic activity of NAG and PH. The activity of SP correlated with pH value (Table 4).

**Table 4.** Correlation between properties of soil on landslide area.

|  | MBN | MBC | MBP | CB | BG | NAG | XYL | SP | PH |
|---|---|---|---|---|---|---|---|---|---|
| C | 0.81 * | 0.79 * | 0.81 * | −0.02 | 0.13 | 0.37 * | 0.29 | −0.12 | 0.47 * |
| N | 0.73 * | 0.82 * | 0.81 * | 0.04 | 0.06 | 0.38 * | 0.23 | −0.09 | 0.58 * |
| pH H$_2$O | −0.25 | −0.29 | −0.40 * | 0.18 | 0.14 | −0.14 | 0.05 | 0.34 * | −0.27 |
| pH KCl | −0.11 | −0.20 | −0.28 | 0.23 | 0.22 | −0.03 | 0.11 | 0.38 * | −0.12 |

* $p < 0.050$.

The GLM analysis confirmed the importance of the deadwood in shaping the carbon content and enzyme activity of soils on the landslide area (Table 5). Additionally, the activity of CB was influenced by localization on the landslide area (zone of depletion or zone of accumulation).

**Table 5.** GLM analysis for carbon content and enzyme activity of soils on landslide area.

| Factors | C | | CB | | BG | | NAG | | XYL | | SP | | PH | |
|---|---|---|---|---|---|---|---|---|---|---|---|---|---|---|
| | *F* | *p* | *F* | *p* | *F* | *p* | *F* | *p* | *F* | *p* | *F* | *p* | *F* | *p* |
| occurrence of deadwood | 5.77 | **0.0215** | 10.06 | **0.0031** | 7.39 | **0.0099** | 9.18 | **0.0044** | 23.71 | **0.0001** | 7.29 | **0.0104** | 7.84 | **0.0081** |
| landslide zone | 2.65 | 0.1118 | 5.53 | **0.0242** | 4.28 | 0.0456 | 0.03 | 0.8586 | 0.80 | 0.3763 | 0.51 | 0.4781 | 1.38 | 0.2473 |
| occurrence of deadwood * landslide zone | 0.24 | 0.6253 | 0.46 | 0.4982 | 0.25 | 0.6187 | 0.22 | 0.7624 | 1.40 | 0.2435 | 0.04 | 0.8402 | 0.21 | 0.6431 |

Significance effect ($p < 0.05$) are shown in bold.

Figure 2 shows the results of the PCA. Factors 1 and 2 explained 57.45% of the variability found in the soil properties, with factor 1 accounting for 33.49% and factor 2 for 23.96%. The first factor is related to the biological activity of soils (enzymatic activity, microbial biomass) and the quantity and quality of soil organic matter (C content, C/N ratio). The second factor is related to the pH of the studied soils and the landslide zone. The soil under the influence of deadwood contained relatively high C and N contents, and exhibited more favorable enzyme activity. Soils without the influence of deadwood were characterized by unfavorable chemical and biochemical properties. The accumulation zone was characterized by a greater stock of deadwood, greater biochemical activity in the soils and higher pH (Figure 2).

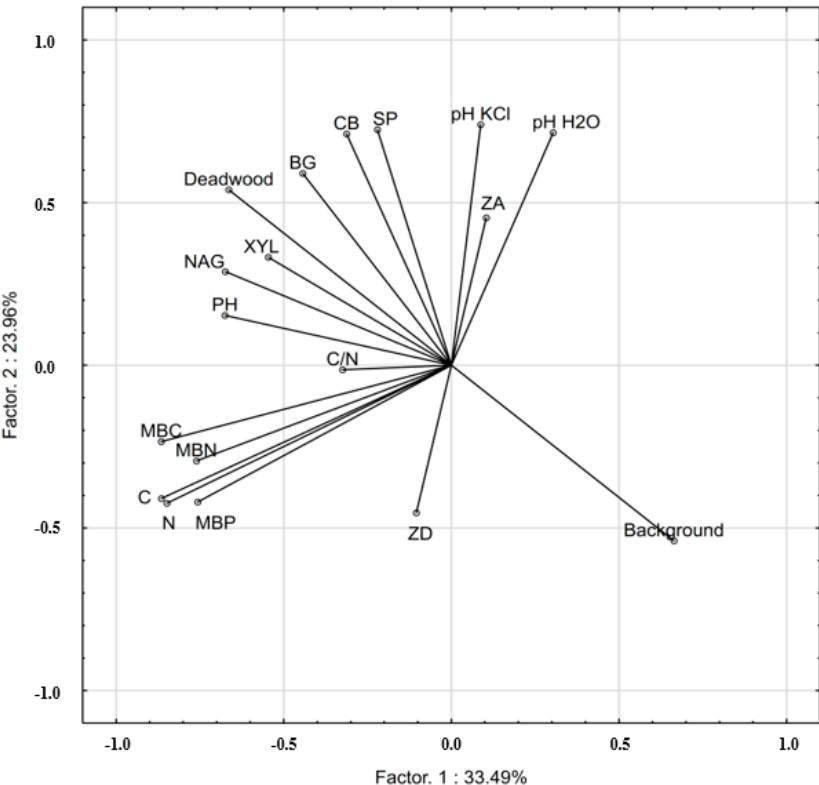

**Figure 2.** Diagram of principal component analysis (PCA) with projection of variables on a plane of the first and second factor (CB—β-D-cellobiosidase, XYL—xylanase, NAG—N-acetyl-β-D-glucosaminidase, BG—β-glucosidase, PH—phosphatase, SP—arylsulphatase; MBC—microbial biomass carbon, MBN—microbial biomass nitrogen (μg·g$^{-1}$), MBP—microbial biomass phosphorus; C—organic carbon content, N—nitrogen content, ZD—zone of depletion, ZA—zone of accumulation).

## 4. Discussion

An overall variation in soil properties was observed across the landslide area, between the depletion and accumulation zones. Soils from the accumulation zone were characterized by a statistically significantly higher amount of deadwood and content of soil organic matter. As part of the landslide process, materials from higher elevations were transported downslope, accumulating at the base of the landslide, resulting in higher soil organic matter content. According to Błońska et al. [9], the depletion zone has been eroded, and is characterized by the least advanced soil-cover recovery. In the lower portion of the landslide, the soil properties result from the mixing of materials from higher elevations and deeper horizons [17]. After 17 years from the occurrence of the landslide, the studied soils are characterized by significant advancement in restoring the soil cover, which is confirmed by the biochemical activity of the soil, expressed by enzymatic activity and microbiological biomass. The soil-cover and vegetation restoration process on landslide areas largely depends on the amount and quality of organic matter available, as this is a nutrient reservoir [9,20,34,35]. The amount and quality of soil organic matter can be used as an indicator of soil condition, as it impacts the physical, chemical and biochemical properties of the soil [36–39]. The microorganisms in the soil influence degradation of organic residue and contribute to the formation of the soil structure [40]. In this study, decomposing deadwood was found to be both a source of organic matter and a source of nutrients essential to the process of soil-cover and vegetation restoration. The concentration of most nutrients in the decaying wood increased during the decay of coarse woody debris, indicating that woody debris retains essential nutrients and that these are gradually released into the soil [41]. The results showed a twofold increase, on average, in the organic C content of soils under the influence of decomposing wood. In the soils of the depletion zone under the influence

of deadwood, we recorded 17.0 g kg$^{-1}$ more carbon compared to the control, and in the soils under the logs in the accumulation zone there was 25.3 g kg$^{-1}$ more carbon. The soil organic C analysis conducted by Pichler et al. [42] showed that differences in concentration in soils with and without coarse woody debris reached 0.04–0.07 kg kg$^{-1}$ in the top 0.5 m layer of soil. Deadwood is gradually incorporated into the soil composition as it releases nutrients and organic C [43–45], and this can significantly influence the restoration of the C balance in the forest ecosystem in a landslide area. In the present study, the majority of dead trees found on the landslide were identified as being in the highest decay classes (III–V). It can be assumed that, within 20 years of the occurrence of an event such as a landslide, with the associated wood decomposition, the soil will be supplied with a considerable amount of C, N and other nutrients that are released throughout the wood decay process. Laiho and Prescott [46] indicate that fir logs after 14 years released almost 30% of their original N content. In our research, we noted a significant influence of the deadwood decomposition process on the N content in soil, especially in the accumulation zone. Decaying wood influenced the content of Ca and Mg in the soil, but it was characterized by large fluctuations. According to Yuan et al. [47], large fluctuations of nutrients may be due to environmental factors that affect the deadwood decomposition.

Landslide areas are reservoirs for significant amounts of deadwood. According to data recorded between 2014 and 2018, the mean lying deadwood volume for more than 30,000 sample plots located all over Poland was 3.9 m$^3$ ha$^{-1}$, with an overall figure for the Carpathian region amounting to 15.2 m$^3$ ha$^{-1}$ [48]. The amount of lying deadwood in landslide areas exceeds this value by about 20 times. This deadwood is a source of various nutrients for soil microorganisms, which process dead organic matter and contribute to its stabilization [12,25]. The deadwood remaining on landslides has a positive impact on soil microbiological activity, which was confirmed by the activity of the tested enzymes and the microbiological biomass of C, N and P. The stimulated enzyme activity is likely an effect of the products being transferred from the deadwood to the soil [11]. The deadwood decomposition process is microbially mediated, and the microorganisms play a major role as decomposition factors [14,15]. On average, the activity of the studied enzymes increased 2-fold in those soils remaining under the influence of substances originating from deadwood. Deadwood is a source of free light fraction soil organic matter, which in turn is an important source of energy for soil microorganisms [49]. On the studied landslide, all analyzed soil enzymes were characterized by significantly higher activity in the soil under the remaining deadwood when compared with the soil from the control sites. The analyzed enzymes participate in the cycling of key nutrients in the soil environment. Compounds such as β-D-cellobiosidase, β-glucosidase and β-xylosidase participate in the decomposition of cellulose. The cellulases are catalogued according to their mechanisms of action as endocellulases (cleaving internal bonds in the cellulose chain), exocellulases or cellobiohydrolases (acting on the reducing or non-reducing ends of cellulose chains) and β-glucosidases (converting cellobiose into glucose molecules) [50]. Cellulose degradation plays a key role in all important nutrient cycles, especially in the carbon cycle [51]. N-acetyl-β-D-glucosaminidase breaks the chemical bonds of glycosides and amino sugars. Phosphatase carries out the decomposition of the esters of phosphoric acid, while arylsulphatase participates in the mineralization of organic sulfur in the soil by carrying out the hydrolysis of sulfate esters. According to Kappes et al. [52], the presence of coarse woody debris enhances soil microbial biomass, the density of detrivores and the species richness of snails, slugs and millipedes. The microbial biomass is related to the content of C and N. According to Peacock et al. [53], the amount and quality of detritus input influence microbial biomass. The soil microbial biomass can be used to indicate changes in the soil organic C content because the soil microbial biomass pool is easily measurable [39]. In the present study, the significance of components released from deadwood on the microbial biomass was confirmed. Statistically significantly higher C and N biomass was recorded in soils under the direct impact of decomposing deadwood. Soil microbial biomass C and N reflect microbial size and soil fertility status, and act as a living nutrient pool in the



soil [54]. The findings of the present study suggest that microbial biomass in soil may be used efficiently to detect long-term changes in soil function, particularly in organic C accumulation. The presence of coarse woody debris enhances the microorganisms and constitutes fluxes of C into the mineral soil [55]. Deadwood plays an important role in the case of poor soil conditions or after disturbance [47].

Regardless of landslide zone, a beneficial effect of decaying wood on soil properties was noted. Decaying wood provides nutrients, especially C substrates, which stimulate the biochemical activity of the soil. On a landslide, where the process of soil-cover restoration takes place, the components released from deadwood may accelerate the processes of soil-cover formation. In this study, it was found that a greater stock of deadwood was present in the accumulation zone, with a greater area of soil surface under its influence; as a result, a stronger effect of the released components on the restoration of soil cover would be expected. The results highlight the contribution of deadwood to sustaining the nutrient status that aids in the regeneration of landslide habitats, particularly those vulnerable to erosion.

## 5. Conclusions

The results of this study emphasize the importance of organic C in acquiring the properties necessary for soil-cover restoration on landslide areas. Deadwood left on the landslide area improves the properties of the soils—-in particular, the amount and quality of soil organic matter and the amount of nutrients. As a result of deadwood decomposition, organic C penetrates the soil, which is a substrate for microbiological reactions. In order to restore landslide soils, microbial activity can be effectively stimulated by leaving deadwood on the landslide surface. The microorganisms found in soil and wood participate in total wood decomposition, which translates into the dynamics and structure of microorganismal communities and, as a consequence, affects the rate of nutrient release, in particular that of C.

**Author Contributions:** Conceptualization, W.P., J.L. and E.B.; data curation, W.P. and E.B.; formal analysis, W.P., J.L. and E.B.; funding acquisition, W.P., J.L. and E.B.; investigation, W.P., J.L. and E.B.; project administration, J.L. and G.G.; visualization, W.P.; writing—original draft, W.P. and E.B.; writing—review and editing, W.P., J.L., G.G. and E.B. All authors have read and agreed to the published version of the manuscript.

**Funding:** This study was financed by the Ministry of Science and Higher Education of the Republic of Poland (BM-2408/2018) and by a subvention from the Ministry of Science and Higher Education of the Republic of Poland for the University of Agriculture in Krakow for 2021.

**Conflicts of Interest:** The authors declare no conflict of interest.

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
