# Peer review of "Effect of Deadwood Decomposition on the Restoration of Soil Cover in Landslide Areas of the Karpaty Mountains, Poland"

_forests, doi:10.3390/f12020237_

Round 1
Reviewer 1 Report
General comments
1) Overall this is an interesting study conducted by Piaszczyk et al., exploring nutrients and microbial soil properties beneath and away from course wood debris following a landslide. I think it is a nice dataset and should be published. However, in the current draft, the authors fall short interpreting the data in away that broader audiences can latch onto. This is essential as I feel the conclusions in this study fall too similar to their previous study Błońska et al. 2018 and Błońska et al 2016 on what may be the same landslide.
2) First, the data is not described in context with plants or trees, so foresters are unable to use this data. Were there trees present? Were plants present near logs or in the background areas? Are the soils bare?
3) Second, the data can be further described in the context of soil C sequestration. There is a nice estimate of the extent of woody debris and I believe the authors can interpret their results into the extent of C from down logs sequestering C in soils, as opposed to entirely going back to the atmosphere as CO2.
4) Lastly, the role of soil formation was poorly investigated. After 17 years of soil formation, is this an average, high, or low degree of soil formation, especially for a landscape prone to landslides? This was not well articulate in this study.
5) In the Material and Method section, the authors state that only the humus horizon (0-10) was sampled. If the authors only studied the organic horizon, then this is a critical study design flaw as the organic horizons as they are directly tied to existing aboveground vegetation. If the authors actually studied A horizons, they need to correct this error. The humus horizon is typically referred to as the organic horizon, according to FAO and US soil taxonomic systems.
Specific comments:
Abstract:
Line 12: Remove ‘effect’
Line 12: The phrase ‘recovering soils in a landslide area’ is unclear. I think the authors need to rephrase to make it clear if the soils are in the depositional area or the source/eroded area.
Line 16: To me, it is incorrect to say positive and negative effect, as they are falsa paradigms in this situation. One can easily say that preventing erosion is negative for generating sediments to rivers for downstream beach formation. Instead, focus if effects are significant and combine with the following sentence on Line 16-17.
Line 18: Be specific about which biochemical parameters.
Lines 20 – 22: Please rephrase. This sentence is unclear.
Introduction:
Line 29: How big is the Carpathian area and is it part of a large geographic province that can be added?
Line 31-38: In addition to flysch rock, what else is promoting landslides in this area? Maybe describe slope, types of soils, geomorphic features present.
Line 45 – 53: I think the authors really need to address re-establishment of plants on the depositional soils. Soil enzymes and microbial communities can be dominated by live plants as opposed to decomposing organic matter. Re-establishment of plants are entirely missing from this introduction. Especially since deadwood is important habitat and germination zone for many tree species.
Line 59: There are been several studies addressing soils following landslides (such as Reddy, V.S. and Singh, J.S., 1993. Changes in vegetation and soil during succession following landslide disturbance in the central Himalaya. Journal of Environmental Management, 39(4), pp.235-250. Which was not cited here), including some of the authors 2018 study (Błońska, E., Lasota, J., Piaszczyk, W., Wiecheć, M. and Klamerus-Iwan, A., 2018. The effect of landslide on soil organic carbon stock and biochemical properties of soil. Journal of Soils and Sediments, 18(8), pp.2727-2737.)
Line 62: The authors should be more explicit in their hypothesis. The one present is vague and does not include how it will be assessed (what would be the control? What parameters are to be studied and why?).
Materials and methods:
Line 69: Please provide physical description of the landslide (how much area and mass?)
Line 85: If only the organic horizon was collected, this likely generated a bias in the study as organic horizons are highly variable spatially due to vegetation (not studied here) and mineral soils are the inorganic material actually controlled by the landslide. Authors need to clarify. If mineral soil sample (A horizons) were sampled, please correct this. Humus horizons are typically referred to as the organic horizon, according to FAO and US soil taxonomic systems.
Line 93: What system was used to described decomposition extent of the deadwood? Please provide citation too.
Figure 1: A scale bar is needed and North-south orientation as well. Topography is also highly warranted.
Line 101: What extraction/digestion was used for base cations?
Line 101: QA/QC are needed for base cation and C and N measurements. Specifically, authors should always report standards used, blanks, and duplicates/triplicates used and their recovery rates. Otherwise, we have no idea of the accuracy of these measurements.
Results:
Line 144: Since base cations were accessed using ICP-OES, there is no need to add the charges as they were not measured even though they are assumed correctly).
Table 3 and 4: Please abbreviate phosphatase as PHO or PHS instead of PH, which can easily be confused with acidity.
Line 200: Please add description of plants in the area. Are there plants in the background area? Are there plants growing near woody debris/logs?
Line 204: To me, the temporal aspect of 17 years of soil formation (2001- 2018) are not interpreted well. There body of literature on early soil formation after disturbance is extensive and poorly used here for soil C accumulation and factors controlling microbial colonization (I am talking about thousands of studies to choose from).
Line 240: Description of how and why the enzymes vary need to be expanded. The authors briefly state that several compounds are associated with cellulose decomposition. Please expand why and how. The literature of cellulose decomposition is poorly used here.
Line 248: No need to say ‘may’ detritus controls microbial communities and biomass.
Line 258: The findings in this study are not clearly interpreted with respect to new or important studies on woody debris, such as:
Laiho, R. and Prescott, C.E., 1999. The contribution of coarse woody debris to carbon, nitrogen, and phosphorus cycles in three Rocky Mountain coniferous forests. Canadian Journal of Forest Research, 29(10), pp.1592-1603.
Magnússon, R.Í., Tietema, A., Cornelissen, J.H., Hefting, M.M. and Kalbitz, K., 2016. Tamm Review: Sequestration of carbon from coarse woody debris in forest soils. Forest Ecology and Management, 377, pp.1-15.
Yuan, J., Hou, L., Wei, X., Shang, Z., Cheng, F. and Zhang, S., 2017. Decay and nutrient dynamics of coarse woody debris in the Qinling Mountains, China. PloS one, 12(4), p.e0175203.
(Please note I am not an author or affiliated with any author on any study suggested in this review).
Overall, this is a nice data set but the interpretations are limited. By comparing these conclusions to the Błońska et al 2018 and 2016, it is clear that the authors are not furthering their data interpretations and run the risk of replicating/duplicating their own previous work. While this is not necessarily a bad thing, these data can be used to draw much broader conclusions about soil C dynamics, soil formation in landslide-prone areas, or plant-soil nutrient cycling.
Author Response
Thank you very much for all the comments to improve the manuscript. Please accept the submission of a revised version of our manuscript. We would like to thank the Reviewers for their diligence and detailed revisions, as well as their insightful and constructive comments. Please find our responses below, showing the changes we have made.
Reviever 1
General comments
Overall this is an interesting study conducted by Piaszczyk et al., exploring nutrients and microbial soil properties beneath and away from course wood debris following a landslide. I think it is a nice dataset and should be published. However, in the current draft, the authors fall short interpreting the data in away that broader audiences can latch onto. This is essential as I feel the conclusions in this study fall too similar to their previous study Błońska et al. 2018 and Błońska et al 2016 on what may be the same landslide.
RESPONSE: Thank you for your valuable comments. We have corrected the manuscript in accordance with the comments
2) First, the data is not described in context with plants or trees, so foresters are unable to use this data. Were there trees present? Were plants present near logs or in the background areas? Are the soils bare?
RESPONSE: The landslide area was without trees, the new generation of fir appeared sporadic, and Rubus hirtus was dominant in undergrowth vegetation. There was no new generation of trees on deadwood and in the immediate vicinity. We added missing information in the text
3) Second, the data can be further described in the context of soil C sequestration. There is a nice estimate of the extent of woody debris and I believe the authors can interpret their results into the extent of C from down logs sequestering C in soils, as opposed to entirely going back to the atmosphere as CO2.
RESPONSE: Thank you for your valuable comments. We have added a few supplementary sentences to the discussion about carbon content in soil under influence of deadwood decomposition process
4) Lastly, the role of soil formation was poorly investigated. After 17 years of soil formation, is this an average, high, or low degree of soil formation, especially for a landscape prone to landslides? This was not well articulate in this study.
RESPONSE: Thank you for your suggestion. We added missing information to the discussion section
5) In the Material and Method section, the authors state that only the humus horizon (0-10) was sampled. If the authors only studied the organic horizon, then this is a critical study design flaw as the organic horizons as they are directly tied to existing aboveground vegetation. If the authors actually studied A horizons, they need to correct this error. The humus horizon is typically referred to as the organic horizon, according to FAO and US soil taxonomic systems.
RESPONSE: Thank you for your suggestion. It was a mistake while editing the text, we changed it
Specific comments:
Abstract:
Line 12: Remove ‘effect’
RESPONSE: It was done
Line 12: The phrase ‘recovering soils in a landslide area’ is unclear. I think the authors need to rephrase to make it clear if the soils are in the depositional area or the source/eroded area.
RESPONSE: It was reformulated
Line 16: To me, it is incorrect to say positive and negative effect, as they are falsa paradigms in this situation. One can easily say that preventing erosion is negative for generating sediments to rivers for downstream beach formation. Instead, focus if effects are significant and combine with the following sentence on Line 16-17.
RESPONSE: It was reformulated
Line 18: Be specific about which biochemical parameters.
RESPONSE: It was reformulated
Lines 20 – 22: Please rephrase. This sentence is unclear.
RESPONSE: It was reformulated
Introduction:
Line 29: How big is the Carpathian area and is it part of a large geographic province that can be added?
RESPONSE: Thank you for your suggestion. We added missing information Carpathian
Line 31-38: In addition to flysch rock, what else is promoting landslides in this area? Maybe describe slope, types of soils, geomorphic features present.
RESPONSE: We added missing information about factors contributing to the formation of landslides
Line 45 – 53: I think the authors really need to address re-establishment of plants on the depositional soils. Soil enzymes and microbial communities can be dominated by live plants as opposed to decomposing organic matter. Re-establishment of plants are entirely missing from this introduction. Especially since deadwood is important habitat and germination zone for many tree species.
RESPONSE: The landslide area was without trees, the new generation of fir appeared sporadic, and Rubus hirtus was dominant in undergrowth vegetation. There was no new generation of trees on deadwood and in the immediate vicinity. We added missing information in the text
Line 59: There are been several studies addressing soils following landslides (such as Reddy, V.S. and Singh, J.S., 1993. Changes in vegetation and soil during succession following landslide disturbance in the central Himalaya. Journal of Environmental Management, 39(4), pp.235-250. Which was not cited here), including some of the authors 2018 study (Błońska, E., Lasota, J., Piaszczyk, W., Wiecheć, M. and Klamerus-Iwan, A., 2018. The effect of landslide on soil organic carbon stock and biochemical properties of soil. Journal of Soils and Sediments, 18(8), pp.2727-2737.)
RESPONSE: It was reformulated and missing refrences were added
Line 62: The authors should be more explicit in their hypothesis. The one present is vague and does not include how it will be assessed (what would be the control? What parameters are to be studied and why?).
RESPONSE: Thank you for your suggestion. The hypotheses have been completed. In our opinion, they reflect the idea of our research
Materials and methods:
Line 69: Please provide physical description of the landslide (how much area and mass?)
RESPONSE: We added missing information about landslide
Line 85: If only the organic horizon was collected, this likely generated a bias in the study as organic horizons are highly variable spatially due to vegetation (not studied here) and mineral soils are the inorganic material actually controlled by the landslide. Authors need to clarify. If mineral soil sample (A horizons) were sampled, please correct this. Humus horizons are typically referred to as the organic horizon, according to FAO and US soil taxonomic systems.
RESPONSE: Thank you for your suggestion. It was a mistake while editing the text, we changed it
Line 93: What system was used to described decomposition extent of the deadwood? Please provide citation too.
RESPONSE: Maser et al. 1979 classification was used and reference was addend to the text
Maser, C.; Anderson, R.G.; Cromack, J.; Kermit Williams, J.T.; Martin, R.E. Dead and Down Woody Material. In Wildlife Habitats in Managed Forests: The Blue Mountains of Oregon and Washington; Jack W.T., Eds.; US Department of Agriculture Forest Service: Washington, DC, USA; 1979, pp. 78–95.
Figure 1: A scale bar is needed and North-south orientation as well. Topography is also highly warranted.
RESPONSE: Figure 1 has been completed in accordance with the comments
Line 101: What extraction/digestion was used for base cations?
RESPONSE: We added missing information
Line 101: QA/QC are needed for base cation and C and N measurements. Specifically, authors should always report standards used, blanks, and duplicates/triplicates used and their recovery rates. Otherwise, we have no idea of the accuracy of these measurements.
RESPONSE: We added missing information
Results:
Line 144: Since base cations were accessed using ICP-OES, there is no need to add the charges as they were not measured even though they are assumed correctly).
RESPONSE: It was removed
Table 3 and 4: Please abbreviate phosphatase as PHO or PHS instead of PH, which can easily be confused with acidity.
RESPONSE: We use the abbreviation PH in all our manuscripts and would prefer not to change it
Line 200: Please add description of plants in the area. Are there plants in the background area? Are there plants growing near woody debris/logs?
RESPONSE: We added missing information about plants to the study area description
Line 204: To me, the temporal aspect of 17 years of soil formation (2001- 2018) are not interpreted well. There body of literature on early soil formation after disturbance is extensive and poorly used here for soil C accumulation and factors controlling microbial colonization (I am talking about thousands of studies to choose from).
RESPONSE: Thank you for your suggestion. We developed a discussion regarding soil formation after 17 years on landslide
Line 240: Description of how and why the enzymes vary need to be expanded. The authors briefly state that several compounds are associated with cellulose decomposition. Please expand why and how. The literature of cellulose decomposition is poorly used here.
RESPONSE: We added missing information
Line 248: No need to say ‘may’ detritus controls microbial communities and biomass.
RESPONSE: It was removed
Line 258: The findings in this study are not clearly interpreted with respect to new or important studies on woody debris, such as: Laiho, R. and Prescott, C.E., 1999. The contribution of coarse woody debris to carbon, nitrogen, and phosphorus cycles in three Rocky Mountain coniferous forests. Canadian Journal of Forest Research, 29(10), pp.1592-1603; Magnússon, R.Í., Tietema, A., Cornelissen, J.H., Hefting, M.M. and Kalbitz, K., 2016. Tamm Review: Sequestration of carbon from coarse woody debris in forest soils. Forest Ecology and Management, 377, pp.1-15; Yuan, J., Hou, L., Wei, X., Shang, Z., Cheng, F. and Zhang, S., 2017. Decay and nutrient dynamics of coarse woody debris in the Qinling Mountains, China. PloS one, 12(4), p.e0175203. (Please note I am not an author or affiliated with any author on any study suggested in this review).
RESPONSE: We added missing references and we developed a discussion section

Reviewer 2 Report
Summary:
This manuscript, by Piaszczyk et al., surveys the deadwood decomposition on landslide restoration. They mainly use microbial biomass C,N and a set of enzyme activity to study the influence of deadwood decomposition in the landslides.
Overall, the topic is of interest to Forests (MDPI) readers, and the methods are sound (for the most part). However, there are some minor issues the way they presented their results.
I suggest some minor revisions before this manuscript is acceptable for publication. I have clearly indicated all my opinions and suggestions in the attached word documents with comments and remarks.
Broad Comments:
Can collecting soil samples from 1m away from the logs be considered background or control soil? Is there a possibility that 20 years of decomposition process would have helped homogenizing the soil adjacent to the logs? Does 1 m be enough to assume that soil wouldn’t have any influence of decomposition. Would be good if the authors add some justification for this sampling technique.
Specific Comments
Methods
It is not clear how the authors measure microbial biomass P. Fumigation-incubation methods is widely used for biomass C and N. Jenkins et al . (1976) reference is for C and N.
Ln 107-109: Please explain why you chose the specific set of enzymes for the enzyme activity analysis.
Results
Table 5: The factors are not clear in the GLM model. What factors were you testing?
Figure 2: What ae factors 1 and 2 in the PCA plots? Are you talking about variation in Axis 1 and Axis 2?
Ln 179-183: How do you incur all this from the vector plot? Please explain.
Ln 215: You did not measure dissolved organic C in your study.
Author Response
Thank you very much for all the comments to improve the manuscript. Please accept the submission of a revised version of our manuscript. We would like to thank the Reviewers for their diligence and detailed revisions, as well as their insightful and constructive comments. Please find our responses below, showing the changes we have made.
Reviever 2
This manuscript, by Piaszczyk et al., surveys the deadwood decomposition on landslide restoration. They mainly use microbial biomass C,N and a set of enzyme activity to study the influence of deadwood decomposition in the landslides. Overall, the topic is of interest to Forests (MDPI) readers, and the methods are sound (for the most part). However, there are some minor issues the way they presented their results. I suggest some minor revisions before this manuscript is acceptable for publication. I have clearly indicated all my opinions and suggestions in the attached word documents with comments and remarks.
RESPONSE: Thank you for the appreciation of our contribution to the research.
Broad Comments:
Can collecting soil samples from 1m away from the logs be considered background or control soil? Is there a possibility that 20 years of decomposition process would have helped homogenizing the soil adjacent to the logs? Does 1 m be enough to assume that soil wouldn’t have any influence of decomposition. Would be good if the authors add some justification for this sampling technique.
RESPONSE: In previous studies, we proved that there was no influence of decaying wood on the soil at a distance of 1 m from the log [Piaszczyk et al. 2019]
Specific Comments
Methods
It is not clear how the authors measure microbial biomass P. Fumigation-incubation methods is widely used for biomass C and N. Jenkins et al . (1976) reference is for C and N.
RESPONSE: Thank you for your suggestion. We added missing information in Methods section
Ln 107-109: Please explain why you chose the specific set of enzymes for the enzyme activity analysis.
RESPONSE: Thank you for your suggestion. We added missing information in introduction
Results
Table 5: The factors are not clear in the GLM model. What factors were you testing?
RESPONSE: Thank you for your suggestion. We added missing information in Statistical analysis subsection
Figure 2: What ae factors 1 and 2 in the PCA plots? Are you talking about variation in Axis 1 and Axis 2?
Ln 179-183: How do you incur all this from the vector plot? Please explain.
RESPONSE: The variables used in the analysis are grouped in directions (axes) on the plane of factors 1 and 2 (Figure 2). The first factor is related to the biological activity of soils (enzymatic activity, microbial biomass) and the quantity and quality of soil organic matter (C content, C / N ratio). The second factor is related to the pH of the studied soils and the landslide zone. We added missing information in the text
Ln 215: You did not measure dissolved organic C in your study.
RESPONSE: It was reformulated
Round 2
Reviewer 1 Report
The reviewers have done a good job addressing my specific concerns.
I feel that the description of the QA/QC is still 'light' the error from standards and deviation among the replicates were not clearly articulated.
One type on Line 265: remove the extra 'the'.
Author Response
The manuscript has been completed in accordance with the comments of the reviewer. Changes have been made to the text.
Reviever 1
Comments and Suggestions for Authors
The reviewers have done a good job addressing my specific concerns.
Thank you for appreciating our work.
I feel that the description of the QA/QC is still 'light' the error from standards and deviation among the replicates were not clearly articulated.
Thanks for your suggestion. We added the missing information:
“All analyzes were performed in the certified Geochemistry Laboratory of Environment and Forest Areas Designed for Reclamation (certificate of Polish Center of Accreditation AB1656). Certified Reference Material (CRM) was used during the all analyzes.”
One type on Line 265: remove the extra 'the'.
It was removed.